# Fatty infiltration in cervical flexors and extensors in patients with degenerative cervical myelopathy using a multi-muscle segmentation model

Monica Paliwal[1]*, Kenneth A. Weber, II[2], Andrew C. Smith[3], James M. Elliott[4,5], Fauziyya Muhammad[1], Nader S. Dahdaleh[6], Jerzy Bodurka[7,8], Yasin Dhaher[9], Todd B. Parrish[10], Sean Mackey[2], Zachary A. Smith[1]

1 Department of Neurosurgery, University of Oklahoma Health Sciences Center, Oklahoma City, Oklahoma, United States of America, 2 Department of Anesthesiology, Systems Neuroscience and Pain Laboratory, Perioperative and Pain Medicine, Stanford University, Palo Alto, California, United States of America, 3 Department of Physical Medicine and Rehabilitation, School of Medicine, Physical Therapy Program, Aurora, Colorado, United States of America, 4 Department of Physical Therapy and Human Movement Sciences, Feinberg School of Medicine, Northwestern University, Chicago, Illinois, United States of America, 5 Faculty of Medicine and Health, University of Sydney, Kolling Institute of Medical Research, St. Leonards, New South Wales, Australia, 6 Department of Neurological Surgery, Feinberg School of Medicine, Northwestern University, Chicago, Illinois, United States of America, 7 Laureate Institute for Brain Research, Tulsa, Oklahoma, United States of America, 8 Stephenson School of Biomedical Engineering, University of Oklahoma, Norman, Oklahoma, United States of America, 9 Department of Physical Medicine and Rehabilitation, University of Texas Southwestern Medical Center, Dallas, Texas, United States of America, 10 Department of Radiology, Feinberg School of Medicine, Northwestern University, Chicago, Illinois, United States of America

* monicapaliwal1@gmail.com

**Data Availability Statement:** All relevant data are within the manuscript and its Supporting information files.

## Abstract

### Background

In patients with degenerative cervical myelopathy (DCM) that have spinal cord compression and sensorimotor deficits, surgical decompression is often performed. However, there is heterogeneity in clinical presentation and post-surgical functional recovery.

### Objectives

Primary: a) to assess differences in muscle fat infiltration (MFI) in patients with DCM versus controls, b) to assess association between MFI and clinical disability. Secondary: to assess association between MFI pre-surgery and post-surgical functional recovery.

### Study design

Cross-sectional case control study.

### Methods

Eighteen patients with DCM (58.6 ± 14.2 years, 10 M/8F) and 25 controls (52.6 ± 11.8 years, 13M/12 F) underwent 3D Dixon fat-water imaging. A convolutional neural network

**Funding:** Zachary A. Smith received funding from the National Institute on Neurological Disorders and Stroke (grant K23NS104211). Kenneth A. Weber II received funding from National Institute of Neurological Disorders and Stroke and National Institute of Child Health and Human Development (grants K23NS104211, L30NS108301 and R03HD094577). Andrew C. Smith received funding from National Institute of Child Health and Human Development (grant R03HD094577). Sean Mackey received funding from National Institute of Drug Abuse (grant K24DA029262). James M. Elliott received funding from National Institute of Child Health and Human Development (grants R01HD079076-01 and R03HD094577). The content is solely the responsibility of the authors and does not necessarily represent the official views of the National Institutes of Health. The funders had no role in the study design, data collection and analysis, decision to publish, or preparation of the manuscript.

**Competing interests:** The authors have declared that no competing interests exist.

(CNN) was used to segment cervical muscles (MFSS- multifidus and semispinalis cervicis, LC- longus capitis/colli) and quantify MFI. Modified Japanese Orthopedic Association (mJOA) and Nurick were collected.

## Results

Patients with DCM had significantly higher MFI in MFSS ($20.63 \pm 5.43$ vs $17.04 \pm 5.24$, $p = 0.043$) and LC ($18.74 \pm 6.7$ vs $13.66 \pm 4.91$, $p = 0.021$) than controls. Patients with increased MFI in LC and MFSS had higher disability (LC: Nurick (Spearman's $\rho = 0.436$, $p = 0.003$) and mJOA ($\rho = -0.399$, $p = 0.008$)). Increased MFI in LC pre-surgery was associated with post-surgical improvement in Nurick ($\rho = -0.664$, $p = 0.026$) and mJOA ($\rho = -0.603$, $p = 0.049$).

## Conclusion

In DCM, increased muscle adiposity is significantly associated with sensorimotor deficits, clinical disability, and functional recovery after surgery. Accurate and time efficient evaluation of fat infiltration in cervical muscles may be conducted through implementation of CNN models.

## Introduction

Degenerative cervical myelopathy (DCM) is a progressive disease that could lead to symptoms such as hyperreflexia, proprioceptive loss, weakness, imbalance, and gait disturbances [1,2]. It is the most common cause of spinal cord dysfunction in the elderly [3,4]. The economic burden of surgical hospitalizations for degenerative cervical spine diseases including DCM is a staggering $USD 2 Billion per annum, not accounting for additional costs owing to time lost from work, rehabilitation costs, and nonsurgical treatment costs [5].

Patients may have varying degrees of cord compression on magnetic resonance imaging (MRI) but it is currently unknown if the severity of compression is directly related to symptom severity [6,7]. Moreover, a considerable number of patients undergoing cervical decompression surgery report less than 50% improvement in clinical function as measured by modified Japanese Orthopedic Association (mJOA) [8]. This variability in clinical presentation and response to surgery indicates a multifactorial nature of DCM's pathophysiology. Several potential pathological mechanisms have been shown to contribute to functional disability in DCM, namely demyelination of spinal cord white matter regions/tracts [9–12], neuronal and volumetric loss of gray matter [13]. Fatty infiltration of cervical spinal musculature is emerging as a potential driver of disability [14]. An improved understanding of specific pathophysiological processes may inform both the clinical assessment of and management for patients with DCM.

Muscle fat infiltration (MFI) is commonly observed in patients with cervical spine diseases including whiplash associated disorders (WAD) from a motor vehicle collision [15], traumatic spinal cord injury [16], and DCM [14]. Increased MFI in cervical flexors (longus capitis/colli and sternocleidomastoid muscles) [17] and extensors (multifidus and semispinalis cervicis) [18,19] is associated with increased pain and clinical disability in patients with WAD. In a smaller clinical cohort of the patients used in this study, we previously showed that in patients with DCM there is an increased MFI in the multifidus muscles that may be associated with clinical disability, as measured by mJOA, and Nurick scales [14]. Changes in muscle

composition may occur due to aging [20], pre-existing degenerative changes [15,21], and/or chronic denervation [22–24]. Insidious damage to the spinal cord in patients with DCM may lead to decreased innervation of the cervical muscles resulting in secondary muscle degeneration, observed as MFI. Along with multifidus muscles, other cervical extensors such as semispinalis cervicis and flexors such as longus capitus and longus colli may also be adversely affected. Therefore, a comprehensive assessment of MFI in cervical spine musculature is warranted to understand both the mechanisms that drive clinical disability and heterogeneity in post-surgical symptoms and recovery. It may help to identify specific muscle groups that are affected and inform interventions through targeted physical therapy (flexors/extensors) regimens to facilitate better cervical neuromuscular function [25,26].

One barrier to quantifying muscle injury is the arduous manual segmentation required of each muscle. Recent advances in the use of artificial intelligence in medical imaging have enabled automated segmentation of cervical muscles [27]. Convolutional neural networks (CNN), in particular, permit a rapid and accurate quantification of MFI.

The purpose of this study was to examine 1) muscle degeneration as MFI of the cervical muscles in patients with DCM compared to healthy controls, 2) associations between MFI and clinical disability; and 3) to demonstrate implementation of a recently developed multi-muscle CNN model to segment seven bilateral cervical muscles and quantify MFI. We hypothesized that patients with DCM will have elevated MFI in deep cervical flexors and extensors and increased MFI will relate to worse myelopathy and clinical dysfunction.

## Material and methods

### Participants

Eighteen patients with DCM (8F/10M, 58.6 ±14.0 years) were recruited from a single academic spine practice. Patients were included if they presented with clinical symptoms of cervical myelopathy such as upper extremity weakness, sensory loss, a lack of hand or leg coordination, or gait instability, in combination with radiographic signs of spinal cord compression. Patients with other neurodegenerative diseases such as Alzheimer's, multiple sclerosis, spinal tumors or trauma, diabetes, peripheral or vascular neuropathies, or a history of spinal injury or other surgery were excluded. Twenty-five age and sex matched healthy controls (12F/13M, 52.6 ± 11.8 years) were recruited. In addition to the exclusion criteria for patients, controls were screened for current spinal conditions, neck pain or other neurological deficits. Apart from these 43 participants, 6 additional participants were excluded from the analysis due to MRI artifacts during image acquisition- misprescribed field of view [4] and fat-water swapping [2]. All participants provided written informed consent as approved by the Northwestern University institutional review board. A subset of these participants was examined in our preliminary study [14].

### Clinical and Health Related Quality of Life Scores (HRQOL)

All participants completed two commonly used clinical scales for myelopathy- mJOA and Nurick. The mJOA scale (ranging from 0–18) assesses sensorimotor dysfunction in upper and lower extremity [28] and the Nurick scale (ranging from 0–4) evaluates ambulatory status [29]. Health related quality of life questionnaires were assessed: Neck Disability Index (NDI), Numerical Rating Scale (NRS) for pain and discomfort, pain interference scale (Pain6a), and Health and Well-Being survey- SF-36 physical (SF-36P) component scores. The NDI evaluates neck pain and its effect on activities of daily living [30]. The NRS scale quantifies pain and discomfort on a scale of 0–10, where 0 = no discomfort/pain, 10 = extreme discomfort/pain [31]. Health and Well-Being survey- SF-36 measures bodily pain, restrictions in physical function and general health due to health problems [32].

A subset of 11 patients with DCM completed these questionnaires at their 6-month follow-up after cervical decompression surgery. In these patients, recovery rate was calculated as (Change in mJOA/(18- Pre-op) *100 [33].

## MRI acquisition and assessment

Participants underwent magnetic resonance imaging of the cervical spine using a 3.0 T Siemens (Erlangen, Germany) Prisma scanner. A 64 channel head [40] and a neck coil [24] were used to acquire high-resolution 3D fat-water images of the cervical and upper thoracic spine (C2-T1) with a dual-echo gradient-echo FLASH sequence (2-point Dixon, TR = 6.59 ms, TE1 = 2.45 ms, TE2 = 3.68 ms, flip angle = 12˚, field-of-view = 190 mm × 320 mm, slab oversampling of 22% with 36 partitions to prevent aliasing in the superior- inferior direction, in-plane resolution = 0.7 mm × 0.7 mm, slice thickness = 3.0 mm, number of averages = 6, acquisition time = 4 min 23 s).

Automated segmentation of the seven bilateral cervical muscle group was performed using a recently trained dense V-net CNN model and the in-phase and out-of-phase images. The model was run using the NiftyNet (Version 0.6.0) open-source deep-learning platform built on TensorFlow (Version 1.15) in Python (Version 3.6) [34,35]. The CNN demonstrated high accuracy and excellent reliability ($ICC_{2,1} > 0.800$) for the MFI measures of all muscle groups in an independent testing dataset. The mean absolute error and root mean squared error in MFI measures was less than 2.0% and 3.0%, respectively. The CNN reduces the time to segment a single dataset from 4 to 8 hours down to only seconds (Weber et. al., in review). MFI was calculated as the fat signal/(fat signal + water signal) × 100 using the mean fat and water signal from each of the muscle segmentations. MFI metrics were extracted for the following cervical muscle groups- left and right multifidus and semispinalis cervicis (MFSS), longus colli and longus capitis (LC), semispinalis capitis (SSCap), splenius capitis (SPCap), levator scapula, (LS), sternocleidomastoid (SCM), and trapezius (TR) [Fig 1]. The average MFI of left and right muscles of each muscle group was used for further analysis.

## Statistical analysis

Normality of the data was assessed using the Shapiro-Wilk test [36]. One-way Analysis of Covariance (ANCOVA) was performed to evaluate significant differences in MFI between subject groups with age, sex, and BMI as covariates. Severity of symptoms was defined using mJOA groups- Normal (mJOA = 18), Mild (mJOA = 17, 16 or 15), Moderate (mJOA = 14, 13, 12) [37]. One-way ANCOVA was conducted to assess significant differences in MFI between mJOA groups and Nurick scores controlling for age, sex and BMI in the model. Spearman's correlation was used to assess the association between clinical or HRQOL scores (mJOA, Nurick, Neck NRS, Arm NRS, NDI, Pain6a, SF-36P) and MFI. Paired sample t-tests were conducted to assess improvement in clinical and HRQOL scores, and Spearman's correlation was used to evaluate the relationship between clinical scores post-surgery and pre-surgical MFI. Statistical tests were performed using IBM SPSS Statistics (IBM Corp. Released 2015. IBM SPSS Statistics for Windows, Version 22.0. Armonk, NY: IBM Corp.) and significance was set at $p \leq 0.05$.

## Results

### Participant characteristics- demographic, clinical & HRQOL scores

The patient cohort consisted of 18 subjects (8 females and 10 males) while the control group consisted of 25 subjects (12 females and 13 males); Female: Male (8F:10M) vs (12F:13M)

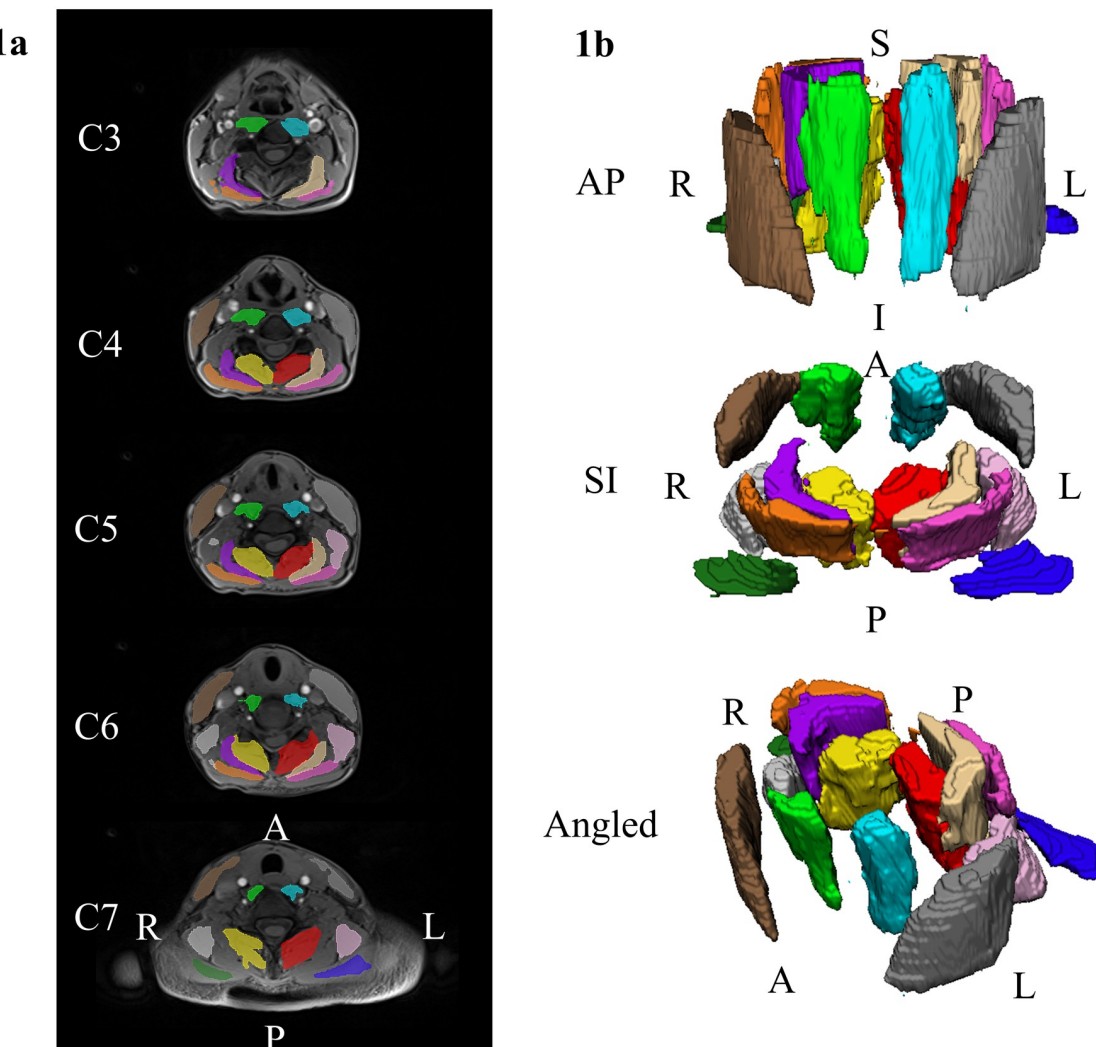

**Fig 1. Example renderings of the cervical spine muscle segmentations.** We used a previously developed convolutional neural network (CNN) to segment seven bilateral cervical muscles (14 muscles total). Example segmentations from a randomly selected DCM participant are shown. a) Two-dimensional renderings of the cervical spine segmentations at the C3 through C7 vertebral levels overlaid the axial water images. b) Three-dimensional renderings of the cervical spine muscle segmentations. The muscle groups segmented included the MFSS (left = red, right = yellow, LC (left = light blue, right = light green), SSCap (left = beige, right = purple), SPCap (left = dark pink, right = orange), LS (left = light pink, right = light gray), SCM (left = dark gray, right = brown), and TR (left = dark blue, right = dark green). L = left, R = right, A = anterior, P = posterior, S = superior, I = inferior. MFSS = multifidus and semispinalis cervicis, LC = longus colli and longus capitis, SSCap = semispinalis capitis, SPCap = splenius capitis, LS = levator scapula, SCM = sternocleidomastoid, TR = trapezius.

($X^2(1)$ = 0.053, p = 0.818), mean ages 58.6 ± 14.2 years vs 52.6 ± 11.8 years (p = 0.144), mean BMI 26.0 ± 4.1 kg/m$^2$ vs 25.2 ± 3.7 kg/m$^2$ (p = 0.523) respectively. Patients had lower mJOA scores (14.7 ± 1.6, ranging from 12–17 vs 18.0 ± 0, p<0.001), higher Nurick (1.8 ± 0.9, ranging from 1–4 vs 0 ± 0, p<0.001), neck discomfort (4.4 ± 1.8 vs 0.2 ± 0.6, p<0.001), arm discomfort (3.6 ± 2.9 vs 0.2 ± 0.4, p<0.001), NDI (16.89 ± 7.50 vs 1.24 ± 2.05, p<0.001) and Pain-6a (61.02 ± 5.53 vs 41.84 ± 3.68, p<0.001); and lower SF-36P (38.87 ± 9.09 vs 56.49 ± 4.30, p<0.001) as compared to controls [Table 1].

**Table 1. Participant's demographic, clinical and HRQOL characteristics.**

| Subjects | Sex | Age (years) | BMI (kg/m²) | mJOA | Nurick | Neck NRS | Arm NRS | NDI | Pain6a | SF-36P |
|---|---|---|---|---|---|---|---|---|---|---|
| **Controls** | 12F 13M | 52.6 (11.8) | 25.2 (3.7) | 18.0 (0.0) | 0.0 (0.0) | 0.2 (0.6) | 0.2 (0.4) | 1.24 (2.05) | 61.02 (5.53) | 56.49 (4.30) |
| **Patients** | 8F 10M | 58.6 (14.2) | 26.0 (4.1) | 14.7 (1.6) | 1.8 (0.9) | 4.4 (1.8) | 3.6 (2.9) | 16.89 (7.50) | 41.84 (3.68) | 38.87 (9.09) |
| **p-value** | 0.818 | 0.144 | 0.523 | <0.001 | <0.001 | <0.001 | <0.001 | <0.001 | <0.001 | <0.001 |

BMI = Body Mass Index, mJOA = modified Japanese Orthopedic Association, NRS = Numerical Rating Scale, Pain6a = Pain Interference Scale, NDI = Neck Disability Index, SF-36P = Health and well-being survey (physical component score).

Mean (SD) and p-value are reported.

## Muscle fat infiltration and its association with clinical and HRQOL scores

Muscle fat infiltration in the deeper cervical muscle groups such as MFSS, and LC differed significantly between patients with DCM and healthy controls. Specifically, patients with DCM had significantly higher MFI in LC as compared to controls (Right LC- 18.57 ± 7.01 vs 13.27 ± 4.74, p = 0.010, Left LC- 18.91 ± 7.38 vs 14.04 ± 5.20, p = 0.023, Mean LC- 18.74 ± 6.7 vs 13.66 ± 4.91; F (1, 38) = 5.81, p = 0.021, partial η2 = 0.133) after controlling for age, sex and BMI (covariates evaluated at age = 55.12, sex = 0.53 and BMI = 25.50) [Fig 2a]. There was a statistically significant between-subject effect of age (F (1, 38) = 17.19, p<0.001, partial η2 = 0.311) and BMI (F(1, 38) = 6.82, p = 0.013, partial η2 = 0.152) but not sex (F(1, 38) = 1.31, p = 0.260, partial η2 = 0.033) on MFI. Similarly, patients with DCM had significantly higher MFI in MFSS as compared to controls (Right MFSS- 20.58 ± 5.84 vs 16.88 ± 5.34, p = 0.046, Left MFSS- 20.69 ± 5.27 vs 17.20 ± 5.31, p = 0.048, Mean MFSS- 20.63 ±5.43 vs 17.04 ± 5.24; F (1, 35) = 1.38, p = 0.043, partial η2 = 0.138) after controlling for age, sex and BMI (covariates evaluated at age = 55.75, sex = 0.50 and BMI = 25.47) [Fig 2b]. There was a statistically significant between-subject effect of age (F (1, 35) = 11.35, p = 0.002, partial η2 = 0.245) and BMI (F (1, 35) = 9.80, p = 0.004, partial η2 = 0.219) but not sex (F (1, 35) = 1.67, p = 0.205, partial η2 = 0.046) on MFI.

There were no significant differences in MFI levels between patients with DCM and controls in SPCap (8.91 ± 4.78 vs 8.05 ± 4.74, (p = 0.562)), SSCap (15.58 ± 5.15 vs 14.22 ± 4.92, (p = 0.385)), SCM (8.70 ± 4.90 vs 6.81 ± 3.88, (p = 0.164)), TR (6.66 ± 4.09 vs 5.45 ± 4.14, (p = 0.344)), and LS (5.35 ± 3.50 vs 4.68 ± 3.48, (p = 0.541)) muscle groups [Table 2].

Increased MFI was significantly associated with clinical disability, pain, and physical dysfunction. Patients with elevated Nurick scores had significantly higher MFI in LC (Spearman's ρ = 0.436 (p = 0.003)), (F (3, 36) = 3.53, p = 0.024, partial η2 = 0.228) [Fig 3a]. Patients with lower mJOA scores had significantly higher MFI in LC (Spearman's ρ = -0.399 (p = 0.008)). Healthy controls (mJOA = 18, MFI- LC = 13.65 ± 4.90) had lower MFI than patients with mild disability (17 ≥ mJOA ≥ 15, MFI- LC = 18.48 ± 7.52) and moderate disability (14 ≥ mJOA ≥12, MFI- LC = 19.0 ± 6.25), (F (2, 37) = 3.60, p = 0.037, partial η2 = 0.163) after adjusting for age, sex and BMI (covariates evaluated at age = 55.12, sex = 0.53, BMI = 25.50) [Fig 3b]. Increased MFI in MFSS were significantly associated with increasing Nurick scores (ρ = 0.341 (p = 0.031)) and decreasing mJOA scores (ρ = -0.332 (p = 0.036)). Similar associations were observed between MFI and HRQOL scores such as NDI (ρ = 0.432 (p = 0.004)), Pain6a (ρ = 0.335 (p = 0.035)), and SF36-P (ρ = -0.420 (p = 0.026)) [Table 3]. Age and BMI had a significant association with MFI in both healthy controls and patients with DCM. Specifically, MFI levels increased with increasing age and BMI in both groups [Fig 4a and 4b].

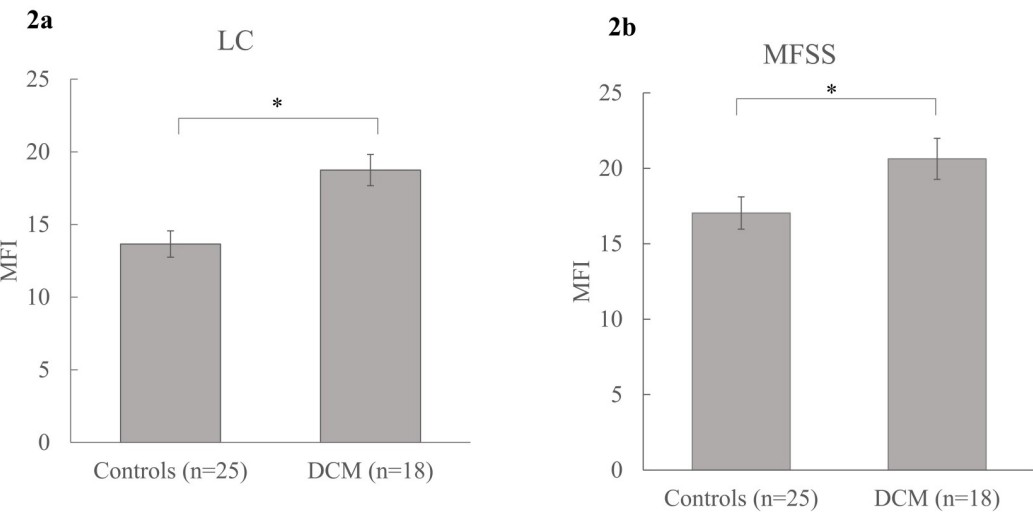

**Fig 2. Mean group differences in MFI.** a) MFI in cervical flexors (LC) between patients with DCM and controls; b) MFI in cervical extensors (MFSS) between patients with DCM and controls. Mean (SE) are reported. * denotes significance at p≤0.05. MFI = muscle fat infiltration, LC = longus colli and longus capitis, MFSS = multifidus and semispinalis cervicis.

## Association with post-surgical recovery

A subset of subjects consisting of 11 patients (4 females and 7 males) with DCM that completed follow up testing after decompression surgery had a mean age of 55.10 ± 15.37 years (ranging from 25–74 years) and BMI of 25.26 ± 3.94 kg/m$^2$. Patients demonstrated significant improvement in their clinical disability after surgery. Nurick scores decreased (0.73 ± 0.65 vs 1.73 ± 0.91, p = 0.004) and mJOA scores increased (14.82 ± 1.90 vs 16.00 ± 1.94, p = 0.029). Higher MFI in LC pre-surgery was associated with post-surgical recovery rate of Nurick (ρ = -0.664 (p = 0.026)) [Fig 5a] and mJOA (ρ = -0.603 (p = 0.049). Similarly, patients who demonstrated ≥50% recovery in mJOA scores had lower MFI (LC) than those with <50% recovery (14.03 ± 7.27 vs 21.09 ±1.28, p = 0.043) [Fig 5b]. However, MFI levels in MFSS before surgery were not associated with post-surgical improvement in Nurick (ρ = 0.139 (p = 0.683)) and mJOA scores (ρ = 0.255 (p = 0.449)).

## Discussion

In this study, we demonstrate that 1a) patients with DCM have higher MFI in deep cervical flexors and extensors as compared to age-and-sex matched healthy controls, 1b) there were no significant group differences in MFI within the more superficial muscles (SPCap, SSCap, SCM, TR, and LS), 2) increased MFI in cervical muscles is associated with higher levels of

**Table 2. Muscle fat infiltration in seven cervical muscle groups.**

|  | MFSS | LC | SPCap | SSCap | SCM | TR | LS |
|---|---|---|---|---|---|---|---|
| **Controls** | 17.04 (5.24) | 13.66 (4.91) | 8.05 (4.74) | 14.22 (4.92) | 6.81 (3.88) | 5.45 (4.14) | 4.68 (3.48) |
| **Patients** | 20.63 (5.43) | 18.74 (6.70) | 8.91 (4.78) | 15.58 (5.15) | 8.70 (4.90) | 6.66 (4.09) | 5.35 (3.50) |
| **p-value** | 0.043 | 0.021 | 0.562 | 0.385 | 0.164 | 0.344 | 0.5S41 |

MFSS = multifidus and semispinalis cervicis, LC = longus colli and longus capitis, SSCap = semispinalis capitis, SPCap = splenius capitis, LS = levator scapula, SCM = sternocleidomastoid, TR = trapezius.

Mean (SD) and p-values are reported.

**3a**

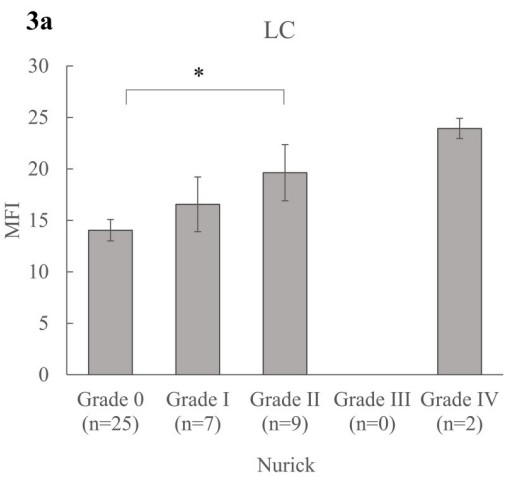

**3b**

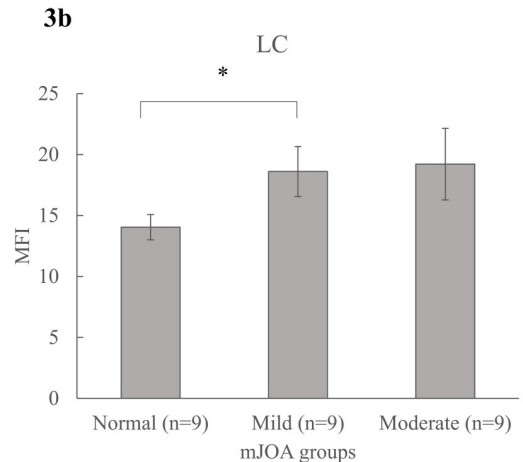

**Fig 3. Association between MFI and clinical scores.** Mean group differences in MFI across varying severity of clinical disability in patients with DCM as measured by a) Nurick (Grade 0—IV), and b) mJOA groups (Normal = 18, Mild = 17–15, and moderate = 14–12). Mean (SE) are reported. * denotes significance at p≤0.05. MFI = muscle fat infiltration, LC = longus colli and longus capitis, mJOA = modified Japanese Orthopedic Association.

clinical disability before and after surgery, and 3) multi-muscle segmentation model, a recently developed deep learning CNN can be applied for automated quantification of MFI in patients with DCM. The findings of this study may be useful in understanding underlying pathological mechanisms that drive injury and clinical dysfunction in DCM. It may also help explain the specific sensorimotor deficits commonly observed in patients with comparable radiographic spinal cord compression [6,7]. MRI findings of altered muscle adiposity are present across a number of spinal pathologies. For example, patients with severe WAD have higher MFI in cervical flexors and extensors as compared to healthy controls and patients with mild/moderate WAD [15–19]. Elevated levels of MFI in chronic whiplash may be reduced through a regimen of neck specific exercises [38]. These changes in muscle composition occur concurrently with decreases in neck disability and increases in muscle strength. In patients with disc degeneration and low back pain, patients with lower MFI in multifidus muscles before total disc replacement surgery had better post-surgical outcomes [39], which is similar to the rotator cuff injury literature [40–42].

These studies suggest that MFI may be an important pathophysiological marker in variety of conditions affecting the appendicular skeleton and the axial spine such as rotator cuff injury [40–42], WAD [15–19], disc herniation [43,44], degenerative disc disease [45], and DCM [14], respectively. However, studies on comprehensive examination of cervical muscle composition and its effects on symptomology, diagnostic, and prognostic utility in patients in DCM are limited.

**Table 3. Association between muscle fat infiltration, clinical and HRQOL scores.**

|  | mJOA | Nurick | NDI | Neck NRS | Arm NRS | Pain6a | SF-36P |
|---|---|---|---|---|---|---|---|
| **MFI (MFSS)** | -0.332 (0.036) | 0.341 (0.031) | 0.346 (0.029) | 0.301 (0.059) | 0.302 (0.060) | 0.335 (0.035) | -0.420 (0.026) |
| **MFI (LC)** | -0.399 (0.008) | 0.436 (0.003) | 0.432 (0.004) | 0.378 (0.012) | 0.420 (0.005) | 0.557 (0.001) | -0.465 (0.008) |

MFI = Muscle Fat Infiltration, MFSS = multifidus and semispinalis cervicis, LC = longus colli and longus capitis, mJOA = modified Japanese Orthopedic Association, NDI = Neck Disability Index, NRS = Numerical Rating Scale, Pain6a = Pain Interference Scale, SF-36P = Health and well-being survey (physical component score). Spearman's ρ (p-value) are reported.

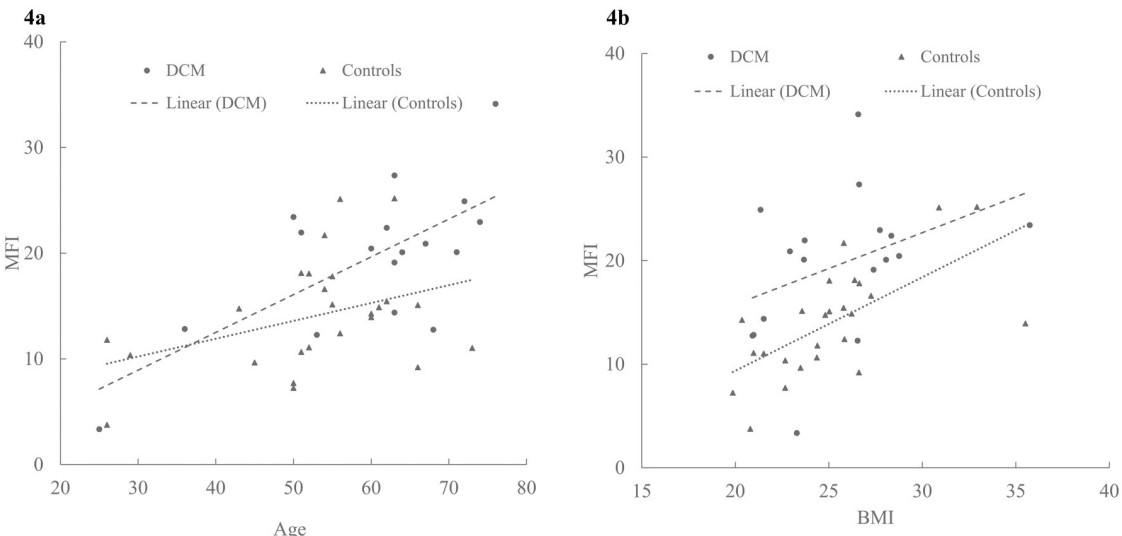

**Fig 4. Association between MFI and demographic characteristics.** Scatterplots depicting the relationship between a) MFI and age in patients with DCM and controls; b) MFI and BMI in patients with DCM and Controls. MFI = muscle fat infiltration, BMI = body mass index.

Changes in muscle composition such as increased MFI of the deep, not superficial, flexors and extensors may have direct implications on the function of the cervical spine muscles and mechanics. Deep neck flexors and extensors provide physical support to the spine vertebral column and play an important role in postural biomechanics, proprioception, and fine motor control [46,47]. Additionally, patients with DCM often present with cervical sagittal vertical misalignment resulting in forward head posture (FHP) [48,49], which could reflect excessive

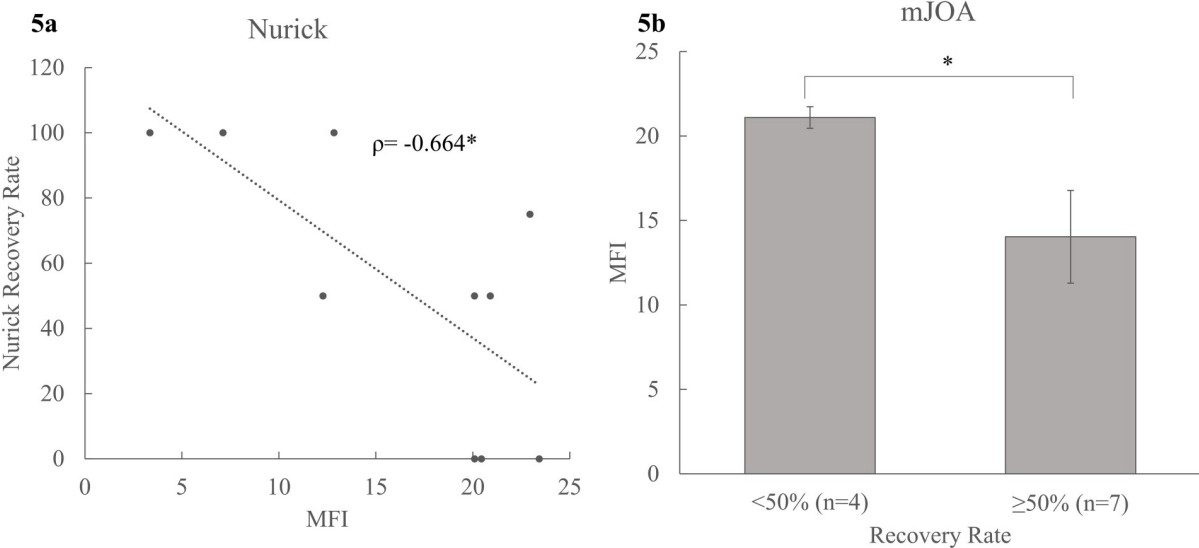

**Fig 5. Association between pre-surgical MFI and post-surgical functional improvement in clinical disability.** a) Scatterplot depicting relation between MFI and Nurick recovery rate, Spearman's ρ is reported; b) Mean group differences in MFI across mJOA recovery rate dichotomized as <50% and ≥50%. Mean (SE) are reported. * denotes significance at p≤0.05. MFI = muscle fat infiltration, LC = longus colli and longus capitis, mJOA = modified Japanese Orthopedic Association.

loading on a weakened muscular system. Biomechanical modeling of increasing FHP and its influence on cervical muscles showed significant lengthening of the cervical extensors such as MFSS and shortening of the cervical flexors such as LC. As a result of sustained contractions, cervical extensor muscles may weaken [50] and could likely fatigue. Prolonged shortening of the cervical flexors may contribute to generalized disuse atrophy and increased MFI [17,51]. Muscle denervation due to spinal cord compression, and the accompanied increased fatty infiltration, may further diminish the capacity of cervical muscles to maintain dynamic posture and tolerate corresponding biomechanical stresses. Sarcopenia refers to loss of muscle mass and function, is a key part of frailty in the elderly [52]. Numerous studies have shown that increased frailty is associated with adverse clinical outcomes after spine surgery in patients with traumatic spinal cord injury [53], degenerative spine diseases [54–56], and DCM [57]. In this study we found that increased MFI is associated with increasing pain and clinical disability as measured by Nurick, mJOA, Neck NRS, and NDI scores. These findings are consistent with the aforementioned literature and validate the impact of muscle composition and quality on the nature/causes of impairment and clinical dysfunction in patients with DCM. In older adults with disability, fatty infiltration in skeletal muscles may be decreased through physical exercise [20,58]. In chronic WAD, MFI in cervical multifidus was reduced after 10 weeks of neck specific exercises [38]. Similarly, neck specific exercises such as flexor/extensor training may be useful in improving clinical outcomes in patients with DCM and MFI may be a modifiable biomarker for therapeutic interventions.

Changes in fat infiltration may occur with increasing age as lean body mass decreases and body adiposity increases [59]. Our findings demonstrate increase in MFI with increasing age and BMI both in healthy controls and patients. However, statistically significant differences in MFI between groups persisted even after controlling for age and BMI. Therefore, we consider our finding of increased MFI in patients with DCM as a pathological and clinically important change in muscle composition.

In patients with DCM, surgical decompression shows variable and limited neurological improvement [60]; surgical interventions combined with conservative rehabilitation show clinical equipoise [61–63]. We previously demonstrated that increased demyelination (lower magnetization transfer ratios (MTR)) in anterior and lateral cord regions and descending motor tracts such as corticospinal, reticulospinal tracts are associated with poor functional recovery after surgery [64]. Sarcopenia in deep cervical flexors (LC) has been shown to predict poor post-surgical improvement in clinical function in patients with myelopathy [65]. In this study, our preliminary analysis evaluating MFI and post-surgical functional recovery showed that increased pre-surgical MFI in LC muscles is adversely related to post-surgical improvement in clinical scores of mJOA and Nurick. It can be hypothesized that cervical flexor training before and after surgery may promote surgical outcomes and MFI could be a predictive biomarker for better prognosis. However, further large sample investigations are needed to understand and confirm the role of cervical musculature in neurological functional recovery and evaluate the utility of fat-water imaging and MFI in predicting surgical outcome in DCM.

Recent advances in artificial intelligence techniques have led to development of automated tools and application of machine learning algorithms in the field of spinal imaging [27,66,67]. In this study, we utilize a recently developed CNN model that segments seven bilateral cervical muscle groups (14 muscles in total) and measures muscle composition in each muscle group (Weber et. al., in review). This CNN was validated in patients with WAD and shown to be highly time efficient as compared to manual segmentation (drawing regions of interest), accurate and reliable for MFI measures (when tested against different human raters). Conventional MRI techniques (T1, T2 weighted imaging) are excellent in detailed visualization of spinal anatomy in spinal diseases such as DCM, however they do not provide information about specific

neuropathophysiological processes such as demyelination, axonal and neuronal loss [68]. Advanced MR techniques such as diffusion tensor, myelin-water, magnetization transfer and fat-water imaging provide several useful quantitative markers of spinal cord injury. Although these MRI based metrics are shown to be associated with clinical impairment and show diagnostic and predictive utility, their translation to clinical use has not been realized [69]. Development and application of CNN models such as the one implemented in this study will be essential to enhance future efficiency and clinical implementation of quantitative MRI techniques.

We acknowledge there are limitations in our study. Firstly, while we have controlled our analysis for confounders such as age, sex, and BMI in assessing differences in MFI; composition of the cervical musculature may also be influenced by other factors such as total lean body mass, physical activity, or neck-specific exercise levels. Secondly, quantification of MFI depends on the accuracy of segmentation of the muscles by the implemented CNN, and subtle differences in field of view during MRI acquisition may affect definition of the muscle boundaries. To address this limitation, we visually screened our database to identify and exclude any such dataset where the CNN may have performed inaccurately. Thirdly, demyelination of the spinal cord regions and white matter tracts may be a prominent pathomechanism in DCM. Here, we studied the role of increased MFI on DCM symptomology in isolation. Additionally, cervical spine misalignment is associated with neck disability in patients with DCM [70], however this study did not control for differences in spinal alignment between participant groups. Future studies that evaluate cumulative impact of cervical spine alignment with cord compression, demyelination, and muscle adiposity are needed as this work shows the early associations that may warrant further investigation. Lastly, while we aimed to include patients with varying degrees of cervical myelopathy, participation of patients with severe myelopathy may be limited due to severity of clinical disability or urgent need of therapeutic interventions.

## Conclusion

Compared to healthy controls, patients with DCM have increased fat infiltration in deep cervical flexors and extensor muscles. These adverse changes in muscle composition associate with sensorimotor deficits, physical function, pain, and disability both before and after surgery. MFI could be a potential biomarker for patient assessments, better candidate selection for surgery and prognosis. Future investigations focusing on predictive utility of fat water imaging should compare differential recovery among patients with DCM after surgery. Secondly, this study demonstrates the novel application of machine learning in medical imaging, specifically in automated segmentation of muscles across the cervical spine and quantification of MRI based metric of muscle composition (MFI). We build upon our previous work [14] utilizing this multi-muscle model that can be easily applied to the cervical spine in DCM and other pathologies. Implementation of machine learning could facilitate translation of MRI metrics such as MFI that are otherwise limited to research environments, into clinical practice.

## Supporting information

**S1 Dataset.**
(XLSX)

## Author Contributions

**Conceptualization:** Zachary A. Smith.

**Data curation:** Monica Paliwal, Nader S. Dahdaleh.

**Formal analysis:** Monica Paliwal, Andrew C. Smith.

**Funding acquisition:** Zachary A. Smith.

**Investigation:** Monica Paliwal, Kenneth A. Weber, II, Andrew C. Smith.

**Methodology:** Monica Paliwal, Kenneth A. Weber, II, Andrew C. Smith, James M. Elliott, Todd B. Parrish, Sean Mackey.

**Resources:** Nader S. Dahdaleh, Todd B. Parrish, Zachary A. Smith.

**Software:** Monica Paliwal, Kenneth A. Weber, II, Andrew C. Smith, Sean Mackey.

**Supervision:** Kenneth A. Weber, II, James M. Elliott, Jerzy Bodurka, Yasin Dhaher, Sean Mackey, Zachary A. Smith.

**Writing – original draft:** Monica Paliwal.

**Writing – review & editing:** Kenneth A. Weber, II, Andrew C. Smith, James M. Elliott, Fauziyya Muhammad, Nader S. Dahdaleh, Jerzy Bodurka, Yasin Dhaher, Todd B. Parrish, Sean Mackey, Zachary A. Smith.

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
