## [Decision Letter · Decision Letter 0]

4 Jun 2021

PONE-D-21-15609

Fatty Infiltration in Cervical Flexors and Extensors in Patients with Degenerative Cervical Myelopathy Using a Multi-Muscle Segmentation Model

PLOS ONE

Dear Dr. Paliwal,

Thank you for submitting your manuscript to PLOS ONE. After careful consideration, we feel that it has merit but does not fully meet PLOS ONE’s publication criteria as it currently stands. Therefore, we invite you to submit a revised version of the manuscript that addresses the points raised during the review process.

We look forward to receiving your revised manuscript.

Kind regards,

Jonathan H Sherman

Academic Editor

PLOS ONE

Journal Requirements:

Reviewers' comments:

Reviewer's Responses to Questions

**Comments to the Author**

1. Is the manuscript technically sound, and do the data support the conclusions?

Reviewer #1: Yes

Reviewer #2: Partly

2. Has the statistical analysis been performed appropriately and rigorously? 

Reviewer #1: I Don't Know

Reviewer #2: Yes

3. Have the authors made all data underlying the findings in their manuscript fully available?

Reviewer #1: Yes

Reviewer #2: Yes

4. Is the manuscript presented in an intelligible fashion and written in standard English?

Reviewer #1: Yes

Reviewer #2: Yes

5. Review Comments to the Author

Reviewer #1: This manuscript looks at fatty infiltration of muscles in the neck in normal patients compared to patients with Cervical myelopathy and tries to assess any associations between amount of fatty infiltration and clinical disability as well as to determine if it predicts post surgical recovery.

They use 18 patients with DCM and 25 controls.

They look at fatty infiltration using MRI sequence with segmentation software

Clinical outcomes were measured with the modified Japanese orthopedic association scoring and Nurick

Introduction: well written and introduces topic clearly. Cleary lays out objectives of the study

Methods: The JoA scores seem high for the DCM group for the most part they seem like high functioning with little symptoms. Is this representative of patients with DCM? Anecdotally I see patients with much lower scores with this pathology and wonder if you would see bigger differences in more severe longstanding pathology.

In that same vein it would be important to determine if length of symptoms and how this factored in with MIF and recovery prediction. The failure of patients to improve may be related to how long the damage was present.

Discussion: Authors make is seem that they believe the MIF is the cause of the lack of clinical improvement. Doesn’t it make more sense that it is a symptom of damage and is merely a predictor of lack of improvement not a cause. In the discussion line 356-358 authors suggest that in addition to demyelination muscle composition may impact resolution of symptoms after therapeutic interventions. I realize they say may but I think that is still a pretty strong statement with no real evidence. I agree it is necessary to explore this question but it think exploration needs to be done before such a statement is made. but i think that statement should be softened a bit more.

Figures are easy to understand.

Also how different is this paper from the 2018 paper looking at MFI in multifidus musculature on the surface it seems very similar. authors should bring up older paper in introduction (if they do i missed it) and add how this advances that data.

Reviewer #2: This is an interesting foundational work in attempting to use machine learning to assess differences in patients with CSM as compared to controls. The authors present their data clearly, but need to avoid any confusion of causation with association, which is not assessed by their study. For example, this statement in the conclusion is not supported: "Differences in MFI may explain the variability in clinical presentation of symptoms, augment assessments, inform interventions, and enhance healthcare received by patients with DCM" While the authors have only assessed CSM vs controls in this early foundational work, this has the potential to be clinically useful and I hope next steps for the authors will include utilizing this methodology to comparatively assess differential recovery amongst patients with CSM after surgery.

6. PLOS authors have the option to publish the peer review history of their article (what does this mean?). If published, this will include your full peer review and any attached files.

Reviewer #1: No

Reviewer #2: No

---

## [Author Response · Author response to Decision Letter 0]

11 Jun 2021

Reviewer #1: This manuscript looks at fatty infiltration of muscles in the neck in normal patients compared to patients with Cervical myelopathy and tries to assess any associations between amount of fatty infiltration and clinical disability as well as to determine if it predicts post surgical recovery.

They use 18 patients with DCM and 25 controls.

They look at fatty infiltration using MRI sequence with segmentation software

Clinical outcomes were measured with the modified Japanese orthopedic association scoring and Nurick

Introduction: well written and introduces topic clearly. Cleary lays out objectives of the study

Methods: The JoA scores seem high for the DCM group for the most part they seem like high functioning with little symptoms. Is this representative of patients with DCM? Anecdotally I see patients with much lower scores with this pathology and wonder if you would see bigger differences in more severe longstanding pathology.

We thank the reviewer for this question. We agree the current group of patients is quite high functioning. We did not exclude severe disease. However, there were instances when patients presented with severe myelopathy and intervention was planned in an expeditious manner. We felt that waiting for an opening on MRI calendar date was likely not in the best interest of the patient. Secondly, many of these patients were also, quite anxious, and less amenable to the study. We know these are quite practical reasons, but we always tried to be respectful of the situation and clinical care. We list this as a limitation of our study in line 393-394. 

Additionally, in patients with severe myelopathy, surgery is often beneficial. However, clinical decision making in the treatment of patients with mild compression is the most variable in clinical practices. With mild disease, there is nuance in understanding the associations between anatomic injury and both symptoms and recovery. We are hoping this is a small step forward in better understanding this group of patients.

In that same vein it would be important to determine if length of symptoms and how this factored in with MIF and recovery prediction. The failure of patients to improve may be related to how long the damage was present.

This is an excellent question. Cervical myelopathy, like many other musculoskeletal diseases, is a result of slowly progressing degenerative processes. While assessing patient histories, there was quite poor recall about the onset of symptoms. Therefore, we do not feel we can appropriately give an estimate of true symptom duration. 

We are currently planning to study patients with mild disease who have are treated with conservative care to assess longitudinal changes in MFI.

Discussion: Authors make is seem that they believe the MIF is the cause of the lack of clinical improvement. Doesn’t it make more sense that it is a symptom of damage and is merely a predictor of lack of improvement not a cause. In the discussion line 356-358 authors suggest that in addition to demyelination muscle composition may impact resolution of symptoms after therapeutic interventions. I realize they say may but I think that is still a pretty strong statement with no real evidence. I agree it is necessary to explore this question but it think exploration needs to be done before such a statement is made. but i think that statement should be softened a bit more.

Figures are easy to understand.

We acknowledge this critique. We agree that MFI is only a predictor, and the current work shows an association with lack of clinical improvement. But we do not believe this work shows causation. We have revised the language reflects this. 

We have removed statements on the impact of MFI on resolution of symptoms. We have modified discussion lines 356-359 as “In this study … pre-surgical MFI in LC muscles is adversely related to post-surgical improvement in clinical scores of mJOA and Nurick. It can be hypothesized that cervical flexor training before and after surgery may promote surgical outcomes …”.

Also how different is this paper from the 2018 paper looking at MFI in multifidus musculature on the surface it seems very similar. authors should bring up older paper in introduction (if they do i missed it) and add how this advances that data.

This work built upon our previous investigation. A small subset of these patients was previously studied (in 2018 paper) with a less robust technique (manual region of interest analysis for the multifidus muscle). We refer to our 2018 paper in the line 81 in the introduction and in line 121 in methods. 

The two primary ways this work advances the data are: 1) we previously did not report on specific muscles. The current work assesses MFI in 7 bilateral cervical muscle groups. Given that different muscles of the neck have unique roles and potential contributions to health and disability, we felt this was a significant new contribution. 2) The current work uses machine learning to identify, segment and quantify MFI in these 7 bilateral muscles in a consistent automated method. We demonstrate the application of this new technical contribution for patients with DCM. However, this technique may be used in future studies to identify both normal and abnormal anatomy and in unique disease states. 

We have added “We build upon our previous work (14) utilizing this multi-muscle model that can be easily applied to the cervical spine in DCM and other pathologies.” at line 409 in conclusion.

Reviewer #2: This is an interesting foundational work in attempting to use machine learning to assess differences in patients with CSM as compared to controls. The authors present their data clearly, but need to avoid any confusion of causation with association, which is not assessed by their study. For example, this statement in the conclusion is not supported: "Differences in MFI may explain the variability in clinical presentation of symptoms, augment assessments, inform interventions, and enhance healthcare received by patients with DCM" While the authors have only assessed CSM vs controls in this early foundational work, this has the potential to be clinically useful and I hope next steps for the authors will include utilizing this methodology to comparatively assess differential recovery amongst patients with CSM after surgery.

We thank the reviewer for their comments. We agree the results show an association and we are not supportive of causation. The specific sentence that was cited by Reviewer #2 has been fully removed and re-written as “MFI could be a potential biomarker for patient assessments, better candidate selection for surgery and prognosis” in line 401. 

Further, we have added “Future investigations focusing on predictive utility of fat water imaging should compare differential recovery among patients with DCM after surgery.” in line 402 to note our future directions in exploring a methodology to assess differential recovery, as noted by this reviewer. This work is an early step for us to build upon.

---

## [Decision Letter · Decision Letter 1]

15 Jun 2021

Fatty Infiltration in Cervical Flexors and Extensors in Patients with Degenerative Cervical Myelopathy Using a Multi-Muscle Segmentation Model

PONE-D-21-15609R1

Dear Dr. Paliwal,

We’re pleased to inform you that your manuscript has been judged scientifically suitable for publication and will be formally accepted for publication once it meets all outstanding technical requirements.

Kind regards,

Jonathan H Sherman

Academic Editor

PLOS ONE

Additional Editor Comments (optional):

Reviewers' comments:

Reviewer's Responses to Questions

**Comments to the Author**

1. If the authors have adequately addressed your comments raised in a previous round of review and you feel that this manuscript is now acceptable for publication, you may indicate that here to bypass the “Comments to the Author” section, enter your conflict of interest statement in the “Confidential to Editor” section, and submit your "Accept" recommendation.

Reviewer #2: All comments have been addressed

2. Is the manuscript technically sound, and do the data support the conclusions?

Reviewer #2: Yes

3. Has the statistical analysis been performed appropriately and rigorously? 

Reviewer #2: Yes

4. Have the authors made all data underlying the findings in their manuscript fully available?

Reviewer #2: Yes

5. Is the manuscript presented in an intelligible fashion and written in standard English?

Reviewer #2: Yes

6. Review Comments to the Author

Reviewer #2: Thanks for the responses and changes to this work. I have no further concerns and look forward to reading future work by this group.

7. PLOS authors have the option to publish the peer review history of their article (what does this mean?). If published, this will include your full peer review and any attached files.

Reviewer #2: No

---

## [Editor Report · Acceptance letter]

17 Jun 2021

PONE-D-21-15609R1 

Fatty Infiltration in Cervical Flexors and Extensors in Patients with Degenerative Cervical Myelopathy Using a Multi-Muscle Segmentation Model  

Dear Dr. Paliwal:

I'm pleased to inform you that your manuscript has been deemed suitable for publication in PLOS ONE. Congratulations! Your manuscript is now with our production department. 

Kind regards, 

on behalf of

Dr. Jonathan H Sherman 

Academic Editor

PLOS ONE